# The Mean ApoC1 Serum Level in Postoperative Samples from Neurosurgical Patients Is Lower than in Preoperative Samples and during Chemotherapy

**DOI:** 10.3390/biology11071021

**Published:** 2022-07-07

**Authors:** Michelle Hilbert, Peter Kuzman, Wolf C. Mueller, Ulf Nestler

**Affiliations:** 1Department of Neurosurgery, University Hospital Leipzig, 04103 Leipzig, Germany; michelle.hilbert@live.de; 2Paul-Flechsig-Institute of Neuropathology, University Hospital Leipzig, 04103 Leipzig, Germany; peter.kuzman@medizin.uni-leipzig.de (P.K.); wolf.mueller@medizin.uni-leipzig.de (W.C.M.)

**Keywords:** apolipoprotein C1, biomarker, glioma, neurosurgery, serum level

## Abstract

**Simple Summary:**

Apolipoprotein C1 is mainly synthesized in the liver and helps in the digestion of fatty acids and in their clearance from the bloodstream. Recent reports have described the role of ApoC1 in malignant tumors of internal organs, exerting an impact on tumor growth and prognosis. Only little is known about ApoC1 in patients with diseases affecting the nervous system, so in this study, samples from neurosurgical patients were examined. A total of 219 samples from 85 patients with brain tumors was compared to 11 samples from patients undergoing non-tumor spine surgery. After drawing the blood sample, it was centrifuged to yield the serum supernatant, which was then assessed by a photometric antibody test (ELISA). The results give the concentration of ApoC1 in the probed blood in µg/mL. From a single sample, it was not possible to determine the underlying disease, such as lumbar disc herniation or brain tumor. Of note, the ApoC1 levels of glioblastoma patients fell significantly after the neurosurgical resection of the tumor and then rose again during chemotherapy. Ongoing research deals with a detailed statistical analysis to determine whether ApoC1 serum levels qualify for use as a blood biomarker for glioblastoma.

**Abstract:**

Serum levels of apolipoprotein ApoC1 have been described in a number of systemic tumor entities as potential biomarkers, but little is known about ApoC1 in neurosurgical patients. A total of 230 serum samples from 96 patients were analyzed using an ELISA technique. Patient diagnoses comprised 70 glioblastomas WHO IV°, 10 anaplastic astrocytomas III°, one anaplastic oligodendroglioma III°, one oligodendroglioma II°, one diffuse astrocytoma II°, one pilocytic astrocytoma I°, and a single case of a spindle cell tumor without WHO grading, as well as 11 spinal interventions. The mean ApoC1 level of the 230 samples was 132.03 µg/mL (median 86.83, SD 292.91). In the 176 glioblastoma samples, the mean ApoC1 level was 130.0 µg/mL (median 86.23, SD 314.9), which was neither different from the whole group nor from patients with spinal interventions (215.1 μg/mL, median 63.6, SD 404.9). In the postoperative samples, the mean ApoC1 level was significantly lower (85.81 μg/mL) than in the preoperative samples (129.64 μg/mL) and in samples obtained during adjuvant chemotherapy (168.44 μg/mL). While absolute ApoC1 serum levels in a patient do not allow for the distinction between neurosurgical histological entities, future analyses will examine whether the time course of ApoC1 in an individual patient can be related to certain treatment stages.

## 1. Introduction

Human ApoC1 belongs to the apolipoprotein C family of proteins, involved in lipid metabolism and constituting a part of HDL and VLDL lipoproteins. ApoC1 inhibits the cholesteryl esther transfer protein and promotes the exchange of cholesterol between the different lipoprotein classes. Apolipoprotein C1 is a protein with 57 amino acids and a molecular weight of 6432 Dalton, encoded on chromosome 19q13.32 [1]. It is mainly synthesized in the liver, and its expression in monocytes is increased during differentiation into macrophages. After immunologic stimulation, ApoC1 was observed to reduce pro-inflammatory cytokine secretion from primary murine microglia and astrocytes, as well as from human macrophages [2]. This immunomodulatory property has prompted ApoC1-related oncologic research, beyond its role concerning cardiovascular risk factors and Alzheimer’s disease [3].

However, despite employing mRNA assessment [4], proteomic examinations [5], and serum level detection [6,7,8], the different studies in solid cancers have not finally answered the question of whether high amounts of ApoC1 expression are associated with better prognosis. Likewise, the expression of ApoC1 in central nervous system tumors has been demonstrated, but little information is available on its prognostic impact [9].

ApoC1 serum levels in neurosurgical patients have not been reported in the literature, though theoretically, ApoC1 from brain tumors can leak through the damaged blood–brain barrier into the bloodstream. To approach this gap in knowledge, we determined ApoC1 serum levels in 230 samples from 96 patients undergoing neurosurgical operations by enzyme-linked immunosorbent assay.

## 2. Materials and Methods

The study protocol and procedures were approved by the local ethics committee (311/19-ek) and all patients signed their informed consent prior to the examination of blood samples. Patient diagnoses are given according to the revised 4th edition of the WHO classification of central nervous system tumors, which was in effect during the sampling period from November 2019 to January 2021 (Table 1). 

The patient group with spinal procedures comprised nine patients with surgery for disc herniation (one cervical) and two patients with posterior lumbar decompression. Post-surgery samples were collected the morning after neurosurgical intervention in the ICU as part of the routine electrolyte monitoring and blood gas analysis. The samples during chemotherapy were obtained in the outpatient clinic when leukocyte and thrombocyte counts were needed before prescribing further chemotherapy. In these patients, according to their clinical status, the time point of sampling varied from 1 to 227 months after surgery (mean 18.7, median 10.0).

The blood samples were centrifuged, and the serum supernatant was collected. After routine diagnostic assessment for liver values and inflammation parameters, the serum specimens were frozen at −20 °C until use. For examination of ApoC1 serum levels, samples were allowed to come to room temperature. The ApoC1-ELISA was performed according to the instructions of the manufacturer (EHAPOC1, Invitrogen, Darmstadt, Germany), in which pipetting of the serum samples into a dilution buffer allows for disintegrating lipoprotein particles and exposing ApoC1 molecules. On a 96-well plate, 20 samples were measured, together with eight standard dilutions in duplicate to generate a four-parameter standard curve. Two dilutions (1:10,000 and 1:50,000) of each sample were assessed in duplicate. After stopping the reaction by sulfuric acid, absorbance was determined twice in a microplate reader with a time interval of about 15 min between measurements (SpectraMax M5, Molecular Devices, Munich, Germany). To correct for inhomogeneities in the wells, background absorbance at 550 nm corresponding to an absorbance minimum of the chromogene tetramethylbenzidine after adding the sulfuric acid was subtracted from assay absorbance at 450 nm. The final result for one sample was calculated as the mean of the obtained eight values and expressed in μg/mL.

Data collection and preparation were performed in Excel (Microsoft), and analysis was performed using SPSS 27 (IBM). Most parameters did not follow a normal distribution, so the non-parametric Mann–Whitney U test and the median test were employed for statistical assessment.

## 3. Results

The detected ApoC1 serum values ranged from 10.76 μg/mL to 4046.91 μg/mL. Mean ApoC1 level in the samples of the whole cohort was 132.03 μg/mL (*n* = 230, SD = 292.91), in glioblastoma samples 130.00 μg/mL (*n* = 176, SD = 314.90), in anaplastic astrocytoma 124.60 μg/mL (*n* = 28, SD = 139.64), and in the samples from patients with spinal procedures 215.10 (*n* = 11, SD = 404.91).

A statistically significant decrease in the mean ApoC1 serum level was found, comparing the postoperative samples with the samples before neurosurgical intervention (Table 2). In the same way, the mean ApoC1 value was significantly higher in the samples obtained during adjuvant chemotherapy compared to postoperative samples. The difference between preoperative samples and samples obtained during chemotherapy did not reach significance. This finding was mostly driven by the values obtained in glioblastoma patients; it did not hold true for further diagnoses.

Lipophilic or hemorrhagic serum was partly but not systematically associated with high ApoC1 absorption values. No refinement of data or removal of extreme values was performed, thus the raw data disclosed large standard deviations and considerable differences between mean value and median (Table 3).

When assessing statistically significant differences by median test in the complete cohort, only the increase in the median ApoC1 level during chemotherapy compared to the median in postoperative samples was confirmed. Neither comparison of presurgical to postsurgical median values in the whole cohort, nor assessment in the glioblastoma patient group yielded statistically significant results (Table 2 and Table 3).

After grouping samples from patients with the same diagnosis, mean ApoC1 serum levels or median levels did not differ significantly between the diagnoses. As shown in the single patients with anaplastic oligodendroglioma, pilocytic astrocytoma, and the (WHO unclassified) spindle cell tumor, the postoperative direction (rise/fall) and the amount of ApoC1 serum level variation were subject to an individual course (Table 2).

## 4. Discussion

The present study examined 230 ApoC1 serum values in a group of 96 neurosurgical patients, comparing a control set of 11 patients with spinal interventions to 85 patients with brain tumors. To the best of our knowledge, a similar cohort has not yet been studied.

As a most interesting result, the mean postoperative ApoC1 values in the whole cohort and in the subset of 70 glioblastoma patients were significantly lower than the mean preoperative values and the mean levels obtained during chemotherapy. In contrast, a single, absolute ApoC1 serum value from a patient allowed neither for a histologic diagnosis nor for the drawing of conclusions about the treatment time point at which the sample was taken.

The obtained ApoC1 serum levels were subject to considerable statistical variation with, in part, large standard deviations. These variations might be explained by different extents of ApoC1 integration into lipoprotein subtypes, together with individual clearing times from the bloodstream differing between patients. In some of the examined patients, hemorrhagic transformation of serum or a lipophilic aspect was associated with higher values. Of note, no refinement of data concerning extreme values had been performed prior to analysis in order to avoid a selection bias. This explains why the assessment of ApoC1 median serum levels by median test only confirmed a significant difference between postoperative samples and chemotherapy samples in the complete study group (Table 3), in contrast to the results for the mean values assessed by the Mann–Whitney U test (Table 2). Ongoing research, therefore, deals with a more sophisticated statistical analysis considering individual-level changes over a certain time span. This analysis is in preparation for the subset of glioblastoma patients.

According to the results presented here, the ELISA technique proved feasible for the assessment of ApoC1 serum levels in neurosurgical patients. Its use had already been described in a similar way with comparable serum levels (around 300 μg/mL) for the detection in patients with gastric cancer, where higher levels had been associated with unfavorable prognosis [8]. Moreover, examination of ApoC1 serum levels in 1239 healthy control subjects using a high-throughput mass-spectrometric technique yielded a median value of 47.0 μg/mL (95% confidence interval 28.3 to 73.3), which is in good concordance with the results presented here [10]. However, monocentric patient recruitment and the low number of non-glioblastoma diagnoses pose some limitations to the technical aspects of the presented study.

Further potential confounding factors influencing ApoC1 serum levels include the BMI of individual patients, underlying hyperlipidemia or vascular diseases, and last but not least, metabolic changes induced by general anesthesia or intraoperative blood loss. These factors will finally play an important role when considering the use of ApoC1 serum levels as a biomarker [7].

It remains unsolved whether higher ApoC1 levels are protective or detrimental in cancer [11]. Our results point to the hypothesis that increased ApoC1 values correspond to a higher tumor burden. The immunomodulatory effect of ApoC1 on macrophages and microglia could help glioma cells to induce an immune escape-like metabolic state. Most recently, this has been underlined by the finding that ApoC1 can reduce the activity of ferroptosis pathways in glioma, thus propagating tumor growth [12].

## 5. Conclusions

In the present cohort of neurosurgical patients, the mean ApoC1 serum level in postoperative samples was lower than in samples before surgery. A detailed statistical analysis of individual ApoC1 values, histological diagnoses, treatment course, and survival time will be mandatory to answer the question of whether ApoC1 serum levels qualify as a biomarker for glioblastoma.

## Figures and Tables

**Table 1 biology-11-01021-t001:** Patient characteristics.

	Number of patients	Mean Age (Years)	Female Gender (%)	Pre Surgery (Samples)	Post Surgery (Samples)	During Chemotherapy (Samples)	Sum Samples
all	96	59.5	46.5	71	68	91	230
glioblastoma	70	63.3	48.9	50	54	72	176
spinal procedure	11	49.6	54.5	11	-	-	11
anaplastic astrocytoma III°	10	44.0	25.0	7	9	12	28
anaplastic oligodendroglioma III°	1	61	female	1	2	-	3
diffuse astrocytoma II°	1	71	male	-	-	3	3
oligodendroglioma II°	1	59	male	-	-	4	4
pilocytic astrocytoma I°	1	20	female	1	2	-	3
not classified	1	37	female	1	1	-	2

**Table 2 biology-11-01021-t002:** Mean ApoC1 serum levels (μg/mL) in the diagnosis groups.

	Number of patients	Pre Surgery	Post Surgery	During Chemotherapy	all
all	96	129.64 ^a^	85.81 ^a,b^	168.44 ^b^	132.03
glioblastoma	70	104.39 ^c^	87.57 ^c,d^	179.61 ^d^	130.00
spinal procedure	11	215.10	-	-	215.10
anaplastic astrocytoma III°	10	191.96	69.58	126.57	124.60
anaplastic oligodendroglioma III°	1	31.25	103.15	-	79.18
diffuse astrocytoma II°	1	-	-	167.78	167.78
oligodendroglioma II°	1	-	-	93.42	93.42
pilocytic astrocytoma I°	1	148.06	88.21	-	108.16
not classified	1	95.62	97.27	-	96.45

^a,b,c,d^ Values with the same superscript letter differ significantly (*p* < 0.05 Mann-Whitney-U test).

**Table 3 biology-11-01021-t003:** Statistical variance of ApoC1 levels (μg/mL).

	Parameter	Pre Surgery	Post Surgery	During Chemotherapy	All
all	mean	129.64	85.81	168.44	132.03
(*n* = 96)	median	90.59	79.04 *	97.29 *	86.83
	SD	177.29	51.25	434.68	292.91
	minimum	10.76	24.61	34.57	10.76
	maximum	1407.96	348.73	4046.91	4046.91
glioblastoma	mean	104.39	87.57	179.61	130.00
(*n* = 70)	median	90.49	77.61	97.29	86.23
	SD	59.41	53.21	485.32	314.90
	minimum	23.69	38.07	34.57	23.69
	maximum	421.01	348.73	4046.91	4046.91
spinal procedure	mean	215.10	-	-	215.10
(*n* = 11)	median	63.64	-	-	63.64
	SD	404.91	-	-	404.91
	minimum	10.76	-	-	10.76
	maximum	1407.96	-	-	1407.96
anaplastic	mean	191.96	69.58	126.57	124.60
astrocytoma III°	median	97.85	45.03	87.79	84.49
(*n* = 10)	SD	198.83	52.90	137.47	139.64
	minimum	74.81	24.61	43.98	24.61
	maximum	623.35	190.00	552.54	623.35

* *p* < 0.05 median test.

## Data Availability

The data presented in this study are available on request from the first author or the corresponding author. The data are not publicly available due to an ongoing doctoral thesis procedure.

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
