# Peer review of "The Mean ApoC1 Serum Level in Postoperative Samples from Neurosurgical Patients Is Lower than in Preoperative Samples and during Chemotherapy"

_biology, 2022, doi:10.3390/biology11071021_

Round 1
Reviewer 1 Report
The manuscript by Michelle Hilbert and colleagues describes the detection of serum levels of apoC1 in neurosurgical patients. The study has been designed and described well and the data are well explained using statistics.
The following issues should, however, be taken into account before publication of the manuscript.
Abstract:
No remarks.
Introduction:
Line 38, please give references to published mRNA, proteomic and serum level approaches.
Line 46, Since the focus of the study is on apoC1, present normally in lipoprotein particles, or at least associated with these vesicles, it would be wise to include a remark on the availability of the protein for interaction in the ELISA. Is it free in solution in some way? What is measured, a part that is on the outside of the vesicles? Maybe you can include a reference or just a sentence on serum presence of the free protein and its accessibility for analysis. Can it in some way influence the later obtained large standard deviations?
Methods:
I understand that the paper is a communication, however, I would give a few details on the ELISA. I know that referring to the instructions is fine, but in this case, since in ELISA’s the way of coating and blocking may be of great relevance, I would include a line on that for convenience (is it measured using monoclonals or polyclonals?). The background subtraction at 550 nm is not directly clear to me (in comparison to regular 450 nm subtraction). Does it have to do with the lipids present in the sample?
Line 59, maybe leave out the remark on avoidance of freeze/thawing.
Line 69, remove ‘windows’ (or instead add Microsoft).
Results:
Table 1: Change column heading ‘female (%)’ to ‘female (%) / gender’ for consistency purposes.
Discussion:
Lines 114 and further, would the presence of the protein in the context of lipoprotein particles be the reason for the statistical variation observed? I appreciate the fact that the authors look for the best statistical analysis in this respect, though.
Line 126, are the apoC1 values comparable, although the type of cancer is different? And, were the samples pre-treated before mass spec?
Just a general question, relating to the remark in line 38/39: the serum levels may hint to a better prognosis, however, do the other studies mention anything about the levels of apoC1 and do they speculate on a biological reason for that? If so, it would be nice to include that in the discussion.
Author Response
Comments of Reviewer 1:
Line 38, please give references to published mRNA, proteomic and serum level approaches.
We have added references by Cohen, Cui and Shi.
Line 46, Since the focus of the study is on apoC1, present normally in lipoprotein particles, or at least associated with these vesicles, it would be wise to include a remark on the availability of the protein for interaction in the ELISA. Is it free in solution in some way? What is measured, a part that is on the outside of the vesicles? Maybe you can include a reference or just a sentence on serum presence of the free protein and its accessibility for analysis. Can it in some way influence the later obtained large standard deviations?
We have added more descriptions: The blood samples were centrifuged and the serum supernatant was collected. - - - The ApoC1-ELISA was performed according to the instructions of the manufacturer (EHAPOC1, Invitrogen) in which pipetting of the serum samples into a dilution buffer allows for disintegrating lipoprotein particles and exposing ApoC1 molecules.
Unfortunately, the manufacturer does neither reveal the components of the dilution buffer, nor the exact nature of the solid-phase antibody and the soluble retrieval antibody. The difficulties with hemorrhagic or lipophilic serum samples are mentioned in the result section and in the discussion. Here we used explicitly the raw data, the publication in preparation is to include removal of statistical extreme values.
Methods: I understand that the paper is a communication, however, I would give a few details on the ELISA. I know that referring to the instructions is fine, but in this case, since in ELISA’s the way of coating and blocking may be of great relevance, I would include a line on that for convenience (is it measured using monoclonals or polyclonals?). The background subtraction at 550 nm is not directly clear to me (in comparison to regular 450 nm subtraction). Does it have to do with the lipids present in the sample?
We added: After stopping the reaction by sulfuric acid, ... - - - To correct for inhomogeneities in the wells, background absorbance at 550 nm corresponding to an absorbance minimum of the chromogene tetramethylbenzidine after adding the sulfuric acid, was subtracted from assay absorbance at 450 nm.
Line 59, maybe leave out the remark on avoidance of freeze/thawing.
The remark has been removed.
Line 69, remove ‘windows’ (or instead add Microsoft).
Microsoft has been put in.
Results: Table 1: Change column heading ‘female (%)’ to ‘female (%) / gender’ for consistency purposes.
Gender was added.
Discussion: Lines 114 and further, would the presence of the protein in the context of lipoprotein particles be the reason for the statistical variation observed? I appreciate the fact that the authors look for the best statistical analysis in this respect, though.
We added: These variations might be explained by different extents of ApoC1 integration into lipoprotein subtypes, together with individual clearing times from the blood stream differing between patients. In some of the examined patients, hemorrhagic transformation of serum or a lipophilic aspect were associated with higher values.
And clarified: This explains why the assessment of ApoC1 median serum levels by median test only confirmed a significant difference between postoperative samples and chemotherapy samples in the complete study group (Table 3), contrasting to the results for the mean values assessed by Mann-Whitney-U test (Table 2).
Line 126, are the apoC1 values comparable, although the type of cancer is different? And, were the samples pre-treated before mass spec?
We added: ... with comparable serum levels (around 300 µg/ml) ...
Mass spectrometry requires a sophisticated preparation of samples, sometimes with precedent enzymatic peptide digestion in order to enable protein identification by the typical spectrum. The detailed description of the method given by Dittrich et al. is somewhat beyond the scope of the draft.
Just a general question, relating to the remark in line 38/39: the serum levels may hint to a better prognosis, however, do the other studies mention anything about the levels of apoC1 and do they speculate on a biological reason for that? If so, it would be nice to include that in the discussion.
Together with two further references, we included: It remains unsolved, whether higher ApoC1 levels are protective or detrimental in cancer [11]. Our results point to the hypothesis that increased ApoC1 values correspond to a higher tumor burden. The immunomodulatory effect of ApoC1 on macrophages and microglia could help glioma cells to induce an immune escape-like metabolic state. Most recently, this has been underlined by the finding, that ApoC1 can reduce the activity of ferroptosis pathways in glioma, thus propagating tumor growth [12].
Reviewer 2 Report
The topic of the paper is an interesting one, regarding the potential use of apoC1 as a biomarker in brain tumors. Among various cancers, glioblastoma is among the ones with the worst prognosis. Therefore, identification of biomarkers is of utmost importance. The rationale of the study is well defined.
A major problem with data interpretation is the fact that apoC1 concentration in plasma is dependent both on that from the systemic circulation and the leakages through BBB, as the authors themselves indicate. Also, numerous reports in the literature indicated that various proteoforms rather than the bulk amount of the protein as assessed by ELISA would be more appropiate to establish correlations. A better approach to generate more reliable results would be to assess the local concentrations would be probably by real time PCR performed on cDNA obtained from RNA extracted from samples obtained via the surgical procedures.
However, as long as pre- and post- surgical apoC1 levels are compared, the study deserves attention and the finding is worthy to be published as a communication.
Author Response
Comments of Reviewer 2:
A major problem with data interpretation is the fact that apoC1 concentration in plasma is dependent both on that from the systemic circulation and the leakages through BBB, as the authors themselves indicate. Also, numerous reports in the literature indicated that various proteoforms rather than the bulk amount of the protein as assessed by ELISA would be more appropiate to establish correlations. A better approach to generate more reliable results would be to assess the local concentrations would be probably by real time PCR performed on cDNA obtained from RNA extracted from samples obtained via the surgical procedures.
Our previous studies [Groll et al., Evangelou et al.] had compared intraoperative samples by histology, MALDI-TOF and qRT-PCR. The results had prompted the idea, to examine the serum values. Unfortunately, single cell RNA or methylome techniques are not yet available in our neurosurgical laboratory, but we share the reviewers view that the presented results are to be confirmed by simultaneous assessment of blood samples and tumor samples, including the molecular techniques.